# The impact of climate change induced alterations of streamflow and stream temperature on the distribution of riparian species

Jennifer B. Rogers[1,2]*, Eric D. Stein[1], Marcus W. Beck[1¤], Richard F. Ambrose[2,3]

1 Biology Department, Southern California Coastal Water Research Project, Costa Mesa, California, United States of America, 2 Environmental Science and Engineering Program, Institute of the Environment and Sustainability, University of California, Los Angeles, California, United States of America, 3 Department of Environmental Health Sciences, Fielding School of Public Health, University of California, Los Angeles, California, United States of America

¤ Current address: Tampa Bay Estuary Program, St. Petersburg, Florida, United States of America
* jbtaylor339@gmail.com

**Data Availability Statement:** The data and analysis scripts are available in this repository https://github.com/JennyTay/FutureBioImpacts_Flow_

## Abstract

Distributions of riparian species will likely shift due to climate change induced alterations in temperature and rainfall patterns, which alter stream habitat. Spatial forecasting of suitable habitat in projected climatic conditions will inform management interventions that support wildlife. Challenges in developing forecasts include the need to consider the large number of riparian species that might respond differently to changing conditions and the need to evaluate the many different characteristics of streamflow and stream temperature that drive species-specific habitat suitability. In particular, in dynamic environments like streams, the short-term temporal resolution of species occurrence and streamflow need to be considered to identify the types of conditions that support various species. To address these challenges, we cluster species based on habitat characteristics to select habitat representatives and we evaluate regional changes in habitat suitability using short-term, temporally explicit metrics that describe the streamflow and stream temperature regime. We use stream-specific environmental predictors rather than climatic variables. Unlike other studies, the stream-specific environmental predictors are generated from the time that species were observed in a particular reach, in addition to long term trends, to evaluate habitat preferences. With species occurrence data from local monitoring surveys and streamflow and stream temperature modeled from downscaled Coupled Model Intercomparison Project - Phase 5 (CMIP5) climate projections, we predict change in habitat suitability at the end-of-century. The relative importance of hydrology and stream temperature varied by cluster. High altitudinal, cold water species' distributions contracted, while lower elevation, warm water species distributions expanded. Modeling with short-term temporally explicit environmental metrics did produce different end-of-century projections than using long-term averages for some of the representative species. These findings can help wildlife managers prioritize conservation efforts, manage streamflow, initiate monitoring of species in vulnerable clusters, and

Temp and archived with Zenodo here: https://zenodo.org/badge/latestdoi/253941508.

**Funding:** Award recipient: SCCWRP This work was funded by the Los Angeles Regional Water Quality Control Board, Region Four under Agreement No. 16-026-140 https://www.waterboards.ca.gov/losangeles/ YES: The funders had input into the study design, NO: but no role in, data collection and analysis, decision to publish, or preparation of the manuscript.

**Competing interests:** The authors have declared that no competing interests exist.

address stressors, such as passage barriers, in areas projected to be suitable in future climate conditions.

## Introduction

Climate change is expected to be a leading cause of species extinction [1, 2]. These concerns are particularly relevant for riparian species because of the combined effect of changing temperature and rainfall patterns, which have led to predictions of both range contraction and expansion for different species [3, 4]. Riparian species will vary in their vulnerability to climate change depending on environmental tolerances, capacity for adaptation, geographic range size, and ability to migrate within the watershed [5]. Management of riparian communities can be more effective with models that predict which species are likely to be vulnerable to changing conditions. For example, species distribution models that reveal trends in habitat availability may initiate proactive conservation projects or monitoring of species in vulnerable habitats even if the species is not currently threatened. However, developing regional projections of species response to climate change is challenging because relationships between stream condition and species' tolerances through difference phases of their life history are rarely known except for well-studied species. The sheer number of streamflow and temperature metrics makes it hard to evaluate which metrics are most critical for habitat suitability [6]. The dynamic nature of both stream conditions and biology, and the presence of microclimates in many regions make it hard to draw conclusions about habitat needs from large-scale or long-term averaged conditions.

The importance of considering the spatial scale of the predictor variables has been investigated in habitat suitability modeling [7, 8], however the temporal scale of predictor variables has not been explored. Species distribution modeling studies that use future projections from climate models use annual or long-term averages of environmental conditions [9–13], but the environmental averages used do not consider the interannual variation in population size or occurrence and how that relates to the interannual variation in environmental conditions. For example, they do not consider that some years the species may be absent entirely, or in other years there may be breeding. In stream habitats, long-term timeseries of streamflow have been used to develop predictive models for habitat suitability because they capture the regional interannual variability [3]. However, attributing long-term averaged streamflow metrics to a species which breeds only after floods, for example, may misdirect efforts to predict future occurrences. In particular, for short-lived aquatic species in climatically extreme areas like regions with Mediterranean climate, populations are highly dynamic in response to recent climate events and long-term averages do not describe species-specific habitat preferences that vary annually. Additionally, considering the short-term streamflow that impacts species occurrence or abundance is especially important because climate models are predicting more intense and frequent extreme precipitation events [14–16] despite a similar interannual average.

There are two aims to this study and six watersheds in southern California are used as a case study. The first is to predict regional habitat suitability trends of species that represent clusters of riparian species that occur longitudinally (headwaters to estuary) and laterally (thalweg to floodplain) in the river network, in response to projected changes in streamflow and stream temperature. A focal species approach [17], which selects species to represent different habitats, is used to evaluate impacts to biological communities that occur in similar niches.

The second aim is to demonstrate an approach for habitat suitability modeling that uses environmental predictors that spatially and temporally align with the species occurrence data and therefore more accurately describe suitable habitat conditions. To accomplish these aims, we model baseline and end-of-century habitat suitability for six focal riparian species with metrics that describe the streamflow and temperature regime. We use regionally downscaled predictions for streamflow and stream temperature from CMIP5 climate modeling that is mapped to the date and location of the species occurrence or absence point, and includes short- and long-term durations of data. We model the probability of end-of-century species occurrence with projected changes in streamflow and stream temperature under climate change scenarios.

## Methods

### Study region

This study focused on natural and semi-natural streams in six major watersheds throughout Los Angeles and Ventura counties in southern California, southwestern United States (Fig 1). Streams in this region drain steep and geologically young mountains to the Pacific Ocean and include perennial, intermittent, and ephemeral flows. The region experiences a Mediterranean climate and receives an average of 400mm of rain annually, almost entirely during the winter months. Stream reach elevation ranges from sea level to 2646m [18] and high elevations experience snow during the winter. Common riparian shrubs and trees include species of willow,

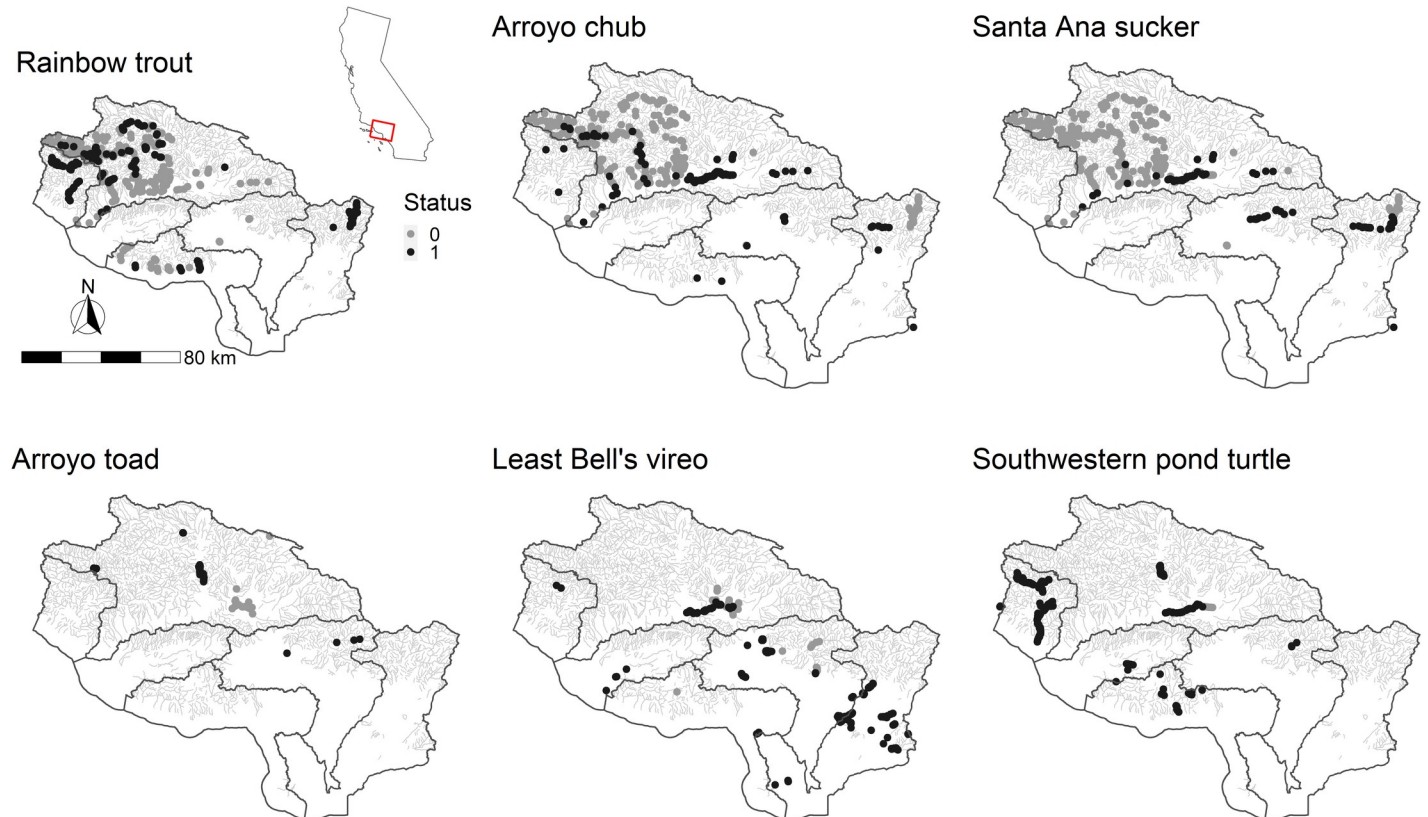

**Fig 1. Study region with inset map showing location within California, USA, in the top left panel.** The flow lines shown are included in this analysis. Blank watershed spaces are areas with heavily altered streams and not included in this analysis. Points represent the observed species data obtained for the study region. Each point represents a unique survey. Grey points denote species absence and black points denote species presence. Note: If a species observation point is shown in areas where there is no flowline, they are occurring along altered streams reaches and not included in this analysis unless a flow gauge is present in that stream reach. Flow line source: USGS national hydrography dataset (NHD, [20]).

Fremont cottonwood, white alder, California sycamore, big-leaf maple, California black walnut, and mulefat [19]. Federally and State endangered and threatened riparian species occur in the region and some are endemic to just a few watersheds. We limited our focus to the mainly unaltered sub-watersheds, much of which exist in the mountains, where stream habitats and floodplains are still intact and changes in climate will be a leading impact on riparian habitat and fauna, rather than land use changes.

## Species selection

We selected species to model using a combined focal species approach [17] and trait-based approach [21]. We selected species that occupy different niches available in the river, longitudinally (headwaters to coastal plain) and laterally (main channel to floodplain), using a clustering approach and selecting focal species from the clusters.

We identified sixty-six riparian vertebrate species in the study area by reviewing regional databases and reports. We included species that require stream or riparian habitat for at least one of their life phases. Through a literature review and input from local experts, we created a life history database that described habitat and behavioral characteristics for each species that will be impacted by climate change, such as stream velocity and vegetation preference (S1 Table in S1 File). We separated the birds from the fish, amphibians, and reptiles because the birds occupied a more terrestrial ecological niche and we did not want that difference to mask the more nuanced differences of stream habitat preference within each group. We transformed the habitat and life history data for both groups from categorical data to a numeric dissimilarity matrix for use in clustering using the function 'daisy' from the package "cluster" [22] in R [23] with the distance metric set to "Gower" (as opposed to Manhattan or Euclidian) which calculates distance between categorical variables based on Gower distance [24]. With this method, species that have identical habitat traits are characterized as identical, and as there are more dissimilar traits between species (for example a preference for fast vs slow velocity), they are classified as more dissimilar from each other. With the dissimilarity matrix, taxa were clustered using a hierarchical clustering method and we cut the dendrogram where dissimilarities were maximized between clusters. There were five clusters of birds and five clusters of fish, amphibians, and reptiles. Final clusters (S2 Table in S1 File) were modified and ultimately two additional clusters were made based on feedback from local experts.

Of the twelve clusters that represented habitat characteristics of riparian species, we selected five clusters to model. We excluded clusters that were composed of entirely invasive species because there is no conservation concern for these clusters. Next, we excluded clusters of habitat generalists, such as dabbling ducks or herons, which use many parts of a stream. While massive changes in water availability would no doubt impact these habitat generalists, subtle changes in streamflow or temperature will have little direct impact, although we do acknowledge that their prey may be impacted by stream changes. Additionally, habitat generalists have been found to relate poorly to focal species [25]. From the five remaining clusters, focal species (Table 1) were selected based on threatened or endangered listing status, the species more dependent on streams than other aquatic habitats, and for modeling purposes, species with reliable occurrence data in the study region. In one cluster, two focal species were selected because they met all the criteria and are both of high management interest. The six focal species in order of geographic range size (largest to smallest) include Southern California steelhead/rainbow trout (*Oncorhynchus mykiss irideus*, trout), Southwestern pond turtle (*Actinemys marmorata*, turtle), Least Bell's vireo (*Vireo bellii pusillus*, vireo) (breeding only), arroyo toad (*Anaxyrus californicus*, toad), arroyo chub (*Gila orcuttii*, chub), and Santa Ana sucker (*Catostomus santaanae*, sucker).

**Table 1. Focal species for habitat modeling.**

| Common name | Scientific name | Life phase | Cluster description | Conservation status | Mean (sd), and max elevation (m) of species occurrence records | Endemism level |
|---|---|---|---|---|---|---|
| Southern California steelhead/ rainbow trout | *Oncorhynchus mykiss irideus* | All | Cool, swift, high gradient streams, coarse substrate, deep pools | FE (Steelhead) | 440(317), 1450 | Low: Alaska through southern CA |
| Southwestern pond turtle | *Actinemys marmorata* | Juvenile / adult | Warm, low to mid gradient stream, deep pools | SSC | 246(168), 711 | Low: Washington to North Baja CA, Mexico |
| Least Bell's vireo | *Vireo bellii pusillus* | Breeding pair | Dense, 5–10 year successional stage, riparian vegetation | FE SE | 173(109), 400 | Medium: Central CA to Baja CA, Mexico |
| Arroyo toad | *Anaxyrus californicus* | Clutch | Temporary shallow backwater pools, sandy substrate | FE SSC | 426(196), 1061 | Medium: San Luis Obispo county to Baja CA, Mexico |
| Arroyo chub | *Gila orcuttii* | All | Warm, sluggish, shallow, backwater or main channel, low to mid gradient streams | SSC | 345(184), 1153 | High: Santa Barbara to San Diego County |
| Santa Ana sucker | *Catostomus santaanae* | All | Warm to cool flowing water, coarse substrate, low to mid gradient stream | FT | 387(162), 808 | High: Los Angeles and Orange Counties |

Conservation status codes: SSC (California Species of Special Concern); FT (federally threatened); FE (federally endangered); ST (state threatened); SE (state endangered). Level of endemism codes: Low (west coast drainages of North America), Medium (central California through Baja California, Mexico), High (Los Angeles region only), CA (California).

## Species distribution data

We compiled the focal species observations within the study region from manuscripts, reports, standardized surveys, memos, and unpublished data sets for the years 1981 through 2017 (Fig 1); sources are shown in S3 in S1 File. A limitation of this approach is the data consisted of disparate sampling efforts and are not distributed randomly across the entire region, however, Soranno [26] found that in many cases combining local data sets can be as good for prediction as data from random sampling. Species observations were only included if location at the National Hydrography Dataset (NHD) [20] reach scale and temporal information at the month scale (or finer) were reported. To minimize pseudo-replication [27], occurrences reported in the same stream reach or the same month were combined into a single data point to ensure that the analysis was not biased toward data rich streams. For example, if a surveyor reported an occurrence every 10m in a stream where they were seining for fish, we assigned that NHD stream reach a single occurrence point, not an occurrence point at each seine location within the reach.

Species occurrence was recorded as presence or absence. If a total count or abundance was provided, a single 'present' point was recorded. Absence was assumed when a surveyor failed to find the species they were looking for or when a surveyor did not record the presence of a species in certain locations but did in others. In some cases, the surveyor was contacted to ensure a lack of species record could be considered an absence rather than a lack of reporting. It is important to recognize that surveys do not achieve perfect detection [28] and thus there is greater uncertainty in the absence data than the presence data.

## Environmental data

This analysis investigated the impact of streamflow and stream temperature on the distribution of species. In Mediterranean regions, like southern California, total streamflow displays large annual variation and most of the precipitation falls in a few storm events [29]. Therefore, the

biologically relevant aspects of the annual hydrograph are better quantified with temporally explicit streamflow metrics that describe attributes such as recession rates or timing of events compared to annual averages [30]. Streamflow metrics were modeled as described in Rogers [31] using gridded precipitation data for three baseline periods terminating in a representative wet (1993), dry (2014), and moderate year (2010), and three predicted end-of-century years: wet (2095), a dry (2090) and a moderate year (2100). Annual stream temperature metrics were also modeled as described in Rogers [31], for baseline years (1982–2014) and end-of-century (2082–2100). The following section describes the approach.

End-of-century streamflow and stream temperature were modeled using climate data from three downscaled global climate models (GCMs) from the Coupled Model Intercomparison Project—Phase 5 (CMIP5) ensemble to consider a range of possible future scenarios assuming the Representative Concentration Pathway (RCP) 8.5 business as usual scenario [32]. The three GCM's include CanESM2, CCSM4, and MIROC5, which were among the ten highest rated models for planning in California based on historical performance and their ability to capture California's climate variability [33].

Daily flow time series were compiled from a combination of flow gages and the U.S. Army Corps of Engineers Hydrologic Engineering Center Hydrologic Modeling System (HEC-HMS) rainfall-runoff model of a subset of watersheds using modeled precipitation data [34]. In total, 68 watersheds were selected to represent our study region and modeled using HEC-HMS with watershed characteristics designated using a method developed by Sengupta [35]. The flow gauges and the downstream terminus of the 68 watersheds were in locations where biological surveys have occurred for the six focal species so that the monitored or modeled flow could be associated with biological condition.

Precipitation for the HEC-HMS models were from a 90-meter, gridded precipitation data set, which consisted of a continuous time series spanning water years 1982–2014 at a 3-hourly time step across the study region [36, 37]. Each HEC-HMS model was run at a 3-hourly time-step and then averaged into daily average flow values, resulting in a daily streamflow time series spanning water years 1982–2014 for each watershed.

To build habitat suitability models (see section below on Biological modeling) hydrologic metrics (S4 Table in S1 File) were calculated from the flow time series (from either the gauge or the model) for each species' presence or absence record at the time and location of the biological survey. Each metric was calculated for short periods from when a species occurrence was observed (3-, 5-, and 10-years prior to the observation) and for the entire duration of the timeseries. To predict habitat suitability throughout the watershed, the hydrologic metrics were estimated regionally (i.e. for every stream reach) using physical basin characteristics and precipitation [34]. Briefly, with the hydrologic metrics from the 68 watersheds and gauged reaches, we used random forest in R [38] to predict streamflow metrics for all NHD reaches in the study for the baseline wet, dry and moderate periods. We used watershed characteristics from the EPA StreamCat database [18], such as elevation and area, and precipitation metrics, such as 'number of storm events', which were derived from the baseline precipitation data.

To predict the flow metrics for the three end-of-century time periods, the random forest model was applied with the same watershed characteristic predictors, but with the end-of-century precipitation data for the wet, dry, and moderate periods. Some watershed characteristics like land cover change can impact stream habitat [39]; however, because our region includes only the unaltered watersheds predominantly in mountainous areas, many of which are protected, we do not anticipate substantial urbanization in future years. All processing for the streamflow metrics was completed in R [40]. Ultimately, we modeled metrics for each NHD stream reach for the baseline and end-of-century years. End-of-century streamflow metrics were averaged across the three GCMs.

Weekly stream temperature maximum, minimum, and mean at the NHD reach scale were modeled using air temperature, watershed elevation (m), watershed area (km$^2$), woody or herbaceous riparian cover (%), and Baseflow Index (%) in a linear regression. The model was trained with local water temperature gauges and gridded air temperature data sets. This model was applied to end-of-century air temperature, which was downscaled from the three GCMs. Six stream temperature metrics (S4 Table in S1 File) were calculated for each year and location with a species observation record, and for all the NHD reaches in the baseline and end-of-century time periods. End-of-century stream temperature metrics were averaged across the three GCMs.

## Biological modeling

We used a species distribution modeling approach [41–43] to predict probability of species occurrence by relating the baseline streamflow and stream temperature metrics to the species occurrence data.

Streamflow metrics at all time frames (3-, 5-, 10-, and all-year) were first tested for the strength of the relationship with each focal species presence or absence using simple logistic regression (S5 Table in S1 File). The metrics that were found to be insignificant (P>0.05, except for the toad where we set the cutoff to P>0.2 because there were so few observations) were removed from the pool of metrics. The remaining streamflow metrics were used to predict species occurrence with a random forest model, which accommodates the large number of streamflow metrics that had significant univariate relationships with each species occurrence. The model was trained with a binary 'presence' or 'absence' outcome, and we converted this to probability of species occurrence by considering the percentage of time that a 'presence' outcome occurred out of the 500 trees that were produced. Validation of the random forest models was done using 25% of the data that had been removed from the training data (Table 2). Model performance is shown in S6 in S1 File. We used the models to predict the probability of occurrence during the baseline and the end-of-century time periods for the wet, dry, and moderate periods.

Logistic regression modeling was used to predict species 'presence' and 'absence' with the stream temperature metrics. We used logistic regression instead of random forest because there were far fewer temperature metrics compared to streamflow metrics. Probability of occurrence was calculated from the log odds (log(p / (1 −p))) by: $P(occurrence)$ = exp(log odds)/(1+exp(log odds)). To minimize collinearity between the six stream temperature metrics, we used principal component analysis (PCA) to reduce the six variables to two uncorrelated predictors, which explained 96% of the variance of the original six metrics (S7 Table in S1 File). We applied the PCA on the standardized baseline and end-of-century stream temperature metrics to calculate the principal components used in the model to calculate species occurrence annually through the end-of-century.

**Table 2. Performance of the random forest model that used the streamflow metrics to predict species distribution.**

|  | Steelhead/ rainbow trout | Southwestern pond turtle | Least Bell's vireo | Arroyo toad | Arroyo chub | Santa Ana sucker |
|---|---|---|---|---|---|---|
| Validation data accuracy (%) | 94 | 97 | 100 | 75 | 93 | 94 |
| Error rate (%) | 7.29 | 5.63 | 6.12 | 32 | 15.73 | 5.91 |
| Number of observations | 566 | 284 | 65 | 33 | 237 | 248 |

The low accuracy and high error rate of the toad is reflective of the low number of observations. The validation data accuracy is the percentage of times the model predicted correctly on the testing dataset. Error rate (out-of-bag estimate of error rate) is the percentage of observations that were misclassified in the training dataset.

## Temporally explicit predictor variables

To investigate the impact of using short-term temporally specific streamflow metrics, versus long-term average streamflow values in habitat suitability modeling, we modeled probability of species occurrence using two additional random forest models. In the first model, we used only the all-year flow metrics in the baseline years to predict probability of occurrence. In the second model (the short-term temporally explicit model), we used only the 3-year metrics in the baseline years to predict the probability of occurrence. We could not do this comparison for the turtle because we only had all-year baseline metrics for that species due to a lack of species observation data within the timespan of the baseline streamflow data. Both models predicted probability of species occurrence by considering the percentage of time that a 'presence' outcome occurred out of the 500 trees that were produced in the random forest, as described above. Models were trained with 75% of the data for each species and were assessed using the error rate and the accuracy on the remaining 25% validation data. We used the two models to predict the probability of occurrence during the end-of-century time period for the wet and dry years. We were interested both in the model performance differences and the differences between the two projected species distribution maps.

## Results

### Environmental drivers

Streamflow variables that relate to stream flashiness such as RBI, recession rates, and flow durations were most associated with the of probability of species occurrence (S5 Table in S1 File). Trout were associated with consistent flows, i.e. there was a negative relationship with storm events and with drought events, and a positive association with hydroperiod. The toad was associated with no-flow periods in the short term (3- and 5-year), but consistent flows in the stream reach over longer time periods (all-year). Chub and sucker were associated with flashiness, rapid recessions, storm events, and were negatively associated with hydroperiod (i.e. they occur in streams that do not flow for part of the year). Chub was generally positively associated with streamflows at all timeframes and magnitudes whereas sucker had negative relationships with flow magnitude at many of the timeframes and magnitude values. Vireo had a positive relationship with flashy streams, fast recessions, and the frequency and recency of storms at all timeframes, but had a negative relationship with hydroperiod. Vireo consistently had a negative relationship with flow magnitude. The turtle was positively related to RBI, yet otherwise was associated with low flow magnitudes, a long duration of low flows, and low storm frequency.

Stream temperature was significantly related to the distribution of five of the six species although the relationships differed among species (Table 3). Relationships observed demonstrate the importance of warm and cool water habitat throughout the region. Because the results of the regression are hard to interpret with the principal components, we show the trend line for probability of occurrence that was predicted from the regression model and each stream temperature metric in S8 in S1 File. Trout show a negative relationship with minimum and mean stream temperatures, yet almost no relationship with maximum stream temperatures. The toad and turtle consistently show a negative relationship with stream temperature for all the metrics. While the regression with the toad did not have significant results, which is expected due to the small sample size compared to the other species, the relationship is supported by the occurrence records which show observations in cooler water compared with the other species (maximum of 29.80˚C, compared to the other species' maximum temperatures, which ranged from 36.44–40.79˚C). Conversely, chub and vireo have a positive relationship

**Table 3. Logistic regression results from the species distribution model driven by stream temperature.**

| Species | Variable | Coefficient (log odds) | Std. error (log odds) | P-value |
|---|---|---|---|---|
| Steelhead/ rainbow trout | PC 1 | 0.05769 | 0.10117 | 0.569 |
| | PC 2 | 0.97616 | 0.22657 | 1.64e-05*** |
| Southwestern pond turtle | PC 1 | -6.8306 | 2.4910 | 0.0061 ** |
| | PC 2 | 2.1587 | 0.9599 | 0.0245 * |
| Least Bell's vireo | PC 1 | 2.86336 | 1.06307 | 0.00707 ** |
| | PC 2 | 0.07816 | 0.51049 | 0.87831 |
| Arroyo toad | PC 1 | -1.6931 | 1.4188 | 0.233 |
| | PC 2 | -0.4166 | 2.8837 | 0.885 |
| Arroyo chub | PC 1 | 0.7959 | 0.1706 | 3.08e-06 *** |
| | PC 2 | -1.1793 | 0.2539 | 3.39e-06 *** |
| Santa Ana sucker | PC 1 | 0.3860 | 0.1788 | 0.0308 * |
| | PC 2 | 0.9909 | 0.5018 | 0.0483 * |

\* P<0.05

\*\* P<0.01

\*\*\* P<0.001. For better interpretation of the coefficients, see S7 in S1 File to see which stream temperature metrics are described by each principal component and see S8 in S1 File, which shows the relationship between the probability of occurrence and each temperature metric implicit in the principal components.

with stream temperatures across all metrics. While vireo is not an aquatic bird, the strong preference for warm water could be mediated through the low elevations that support their preferred riparian vegetation for nesting, their insect prey relationship with stream temperature, or some other factor associated with warm water. Sucker had a positive relationship with maximum and mean stream temperatures but a negative relationship with minimum temperatures.

### Projected changes in species occurrence due to stream temperature

Regional probability of occurrence increased annually for two species and decreased annually for four species from the baseline to the end-of-century (Fig 2). By end-of-century, probability of occurrence increased most rapidly for chub ($\beta_0 = 0.002$, $R^2 = 0.71$), followed by vireo ($\beta_0 = 0.001$, $R^2 = 0.70$). Probability of occurrence decreased for turtle ($\beta_0 = -0.002$, $R^2 = 0.70$), trout ($\beta_0 = -0.001$, $R^2 = 0.61$), sucker ($\beta_0 = -0.001$, $R^2 = 0.45$), and the toad ($\beta_0 = -0.0002$, $R^2 = 0.69$).

We predict that species distributions will shift between high and low elevations under future climate due to stream temperature (Fig 3). The two species that increased overall in probability due to stream temperature increased more in the high elevation streams compared to the low elevation streams. Probably of occurrence increased more for chub at high vs. low elevations (mean probability increase of 0.21 in high elevations vs. 0.09 in low elevations; P< 0.001). Similarly, probability of occurrence for vireo increased by a mean of 0.05 in low elevations vs. mean of 0.13 in high elevations (P< 0.001). In contrast, probability of occurrence for sucker decreased by a mean of -0.11 in high elevations vs. -0.03 in low elevations (P< 0.001). The probability of toad occurrence decreased by a mean of -0.02 in high elevations compared to 0.00 in low elevations (P< 0.001). While the mean decrease in high elevations appears minimal, there are two regions where probability of toad occurrence decreased more substantially (Fig 3 panel C). Probability of trout occurrence decreased by similar magnitudes in high and low elevations (a mean of -0.11 in high elevations and -0.12 in low elevations (P = 0.003)). Finally, the probability of turtle occurrence decreased by a mean of -0.16 in high elevations and -0.23 in low elevations (P< 0.001).

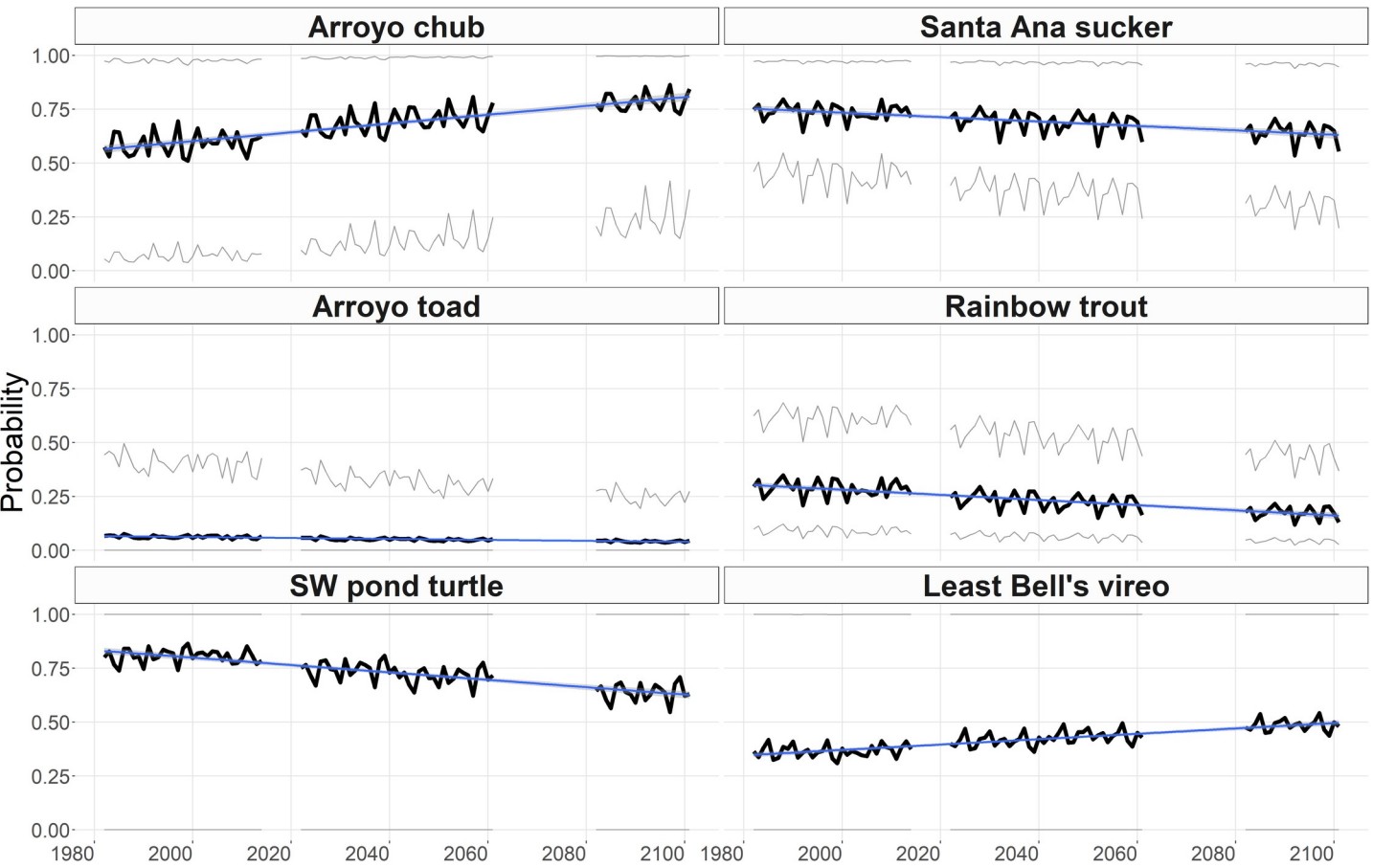

**Fig 2. Annual predicted probability of occurrence from the baseline years to the end-of-century due to stream temperature changes for each species.** The dark black line shows the mean probability of occurrence annually across all the stream reaches. The upper and lower light grey lines show the 95th and 5th percentile probabilities for the stream reaches. Breaks in line show the years that were missing from our analysis. The trend line is a linear model of the mean annual probability of occurrence.

### Projected changes in species occurrence due to streamflow

A positive change in probability of occurrence was generally observed in the wet, moderate, and dry years for the three fish species (Table 4). Conversely, there was a negative change for all three year types for vireo. The probability of occurrence of toad had a mean increase in the dry and moderate year, but a mean decrease in the wet year. The turtle showed no change in probability of occurrence because in both time periods there were predicted probabilities of 1.0.

The elevation of the stream reach did impact the change in probability of occurrence in the end-of-century compared to baseline (Fig 4). The effect was manifested in two ways. First, there was a shift in the same direction but of different magnitude, as seen with chub, trout, and vireo. Second, there was a shift in the opposite direction depending on elevation. For example, probability of occurrence decreased for sucker in low elevations but increased in high elevations. The same was observed for the toad.

### Impacts of stream temperature versus streamflow

We estimate that the change probability of occurrence in end-of-century compared to baseline is likely going to be driven more by changes in stream temperature than streamflow for the

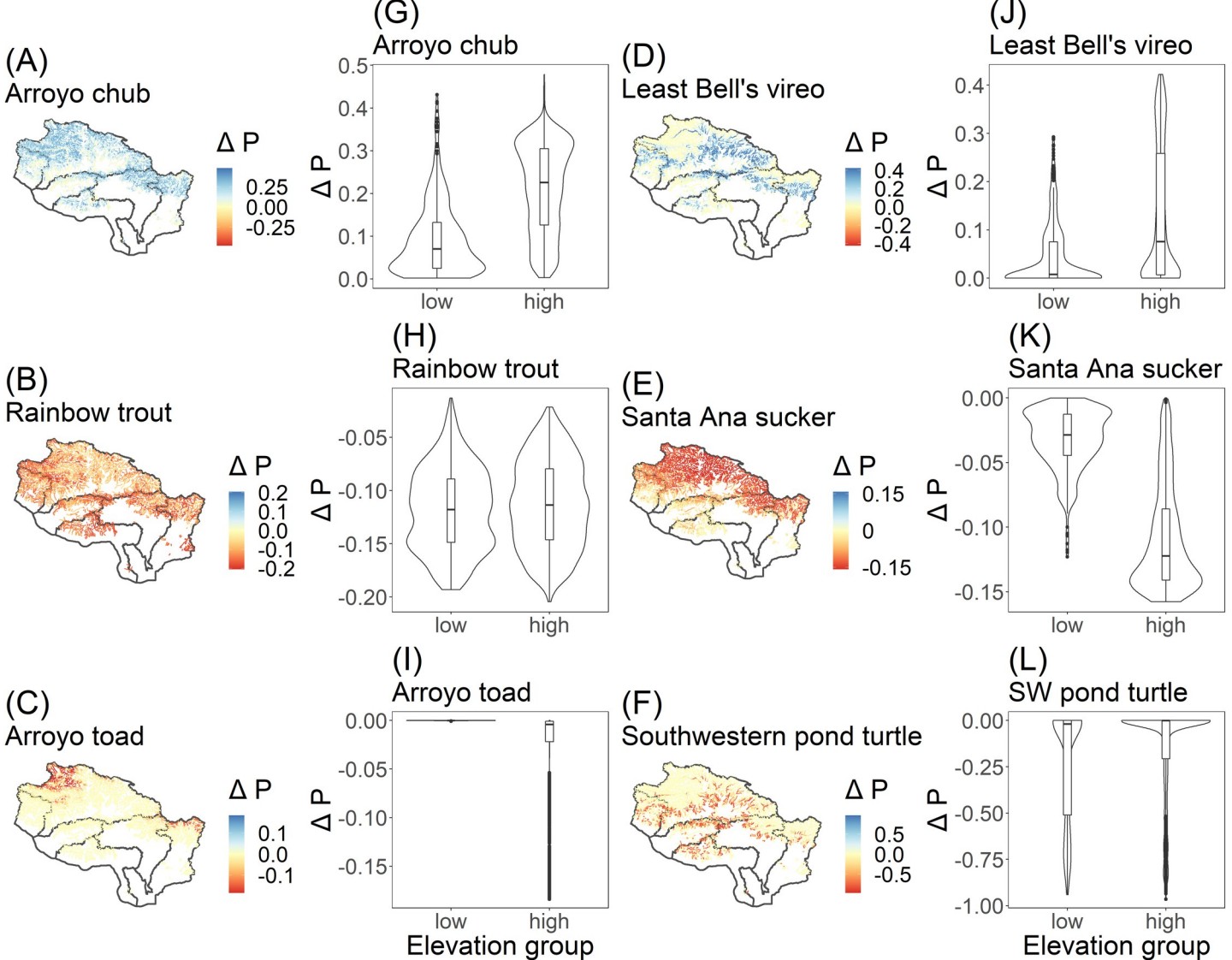

**Fig 3. Change in probability of occurrence due to stream temperature.** Change was calculated as the average values for each stream reach in the end-of-century minus the average values for each stream reach in the baseline period. A-F show the spatial distribution of the change. Note that between species, the color scale is the same (i.e. blue is a positive change and red is a negative change), but the range of values is different. G-L show the distribution of the change in probabilities for the stream reaches between high and low elevations, which is defined as 375m and higher. Note that the y-axis limits are different for each species.

turtle and vireo, but the other four species will experience a similar change in habitat suitability due to each variable (Fig 5). Averaging across year types the average change in probability of occurrence of turtle due to flow was 0.00 and due to temperature was -0.17 (P< 0.001). Similarly, the average change in vireo due to flow was -0.06 and due to temperature was 0.13 (P< 0.001). Alternatively, the average magnitude of the change in probability of occurrence of chub was more similar across the two variables, 0.15 due to flow vs. 0.18 due to temperature (P< 0.001). Likewise, the average change due to streamflow and temperature was 0.06 and -0.07 (sucker, P< 0.001); 0.05 and -0.03 (toad, P< 0.001); 0.10 and -0.10 (trout, P< 0.001), respectively.

The limiting variable stayed the same in the baseline and end-of-century for four species and changed for two. For example, in the baseline, temperature was limiting for the toad and

**Table 4. The average change probability of occurrence throughout the region due to streamflow metrics from the baseline years to the end-of-century years.**

|  | Dry | Moderate | Wet |
|---|---|---|---|
| Arroyo chub | 0.17 (-0.05,0.39) | 0.16 (-0.05,0.34) | 0.11 (-0.04,0.25) |
| Santa Ana sucker | 0.08 (-0.11,0.31) | 0.05 (-0.14,0.23) | 0.07 (-0.07,0.24) |
| Steelhead/ Rainbow trout | 0.11 (-0.05,0.29) | 0.07 (-0.16,0.3) | 0.12 (-0.07,0.33) |
| Least Bell's vireo | -0.08 (-0.2,0.04) | -0.06 (-0.18,0.06) | -0.03 (-0.12,0.08) |
| Arroyo toad | 0.13 (-0.06,0.34) | 0.04 (-0.07,0.18) | -0.02 (-0.12,0.13) |
| SW pond turtle | 0 (0,0) | 0 (0,0) | 0 (0,0) |

Change calculated as probability of occurrence in the end-of-century minus probability of occurrence in the baseline. Numbers in parentheses show the 5th and 95th percentile.

temperature continued to be limiting in the end-of-century. Conversely, in the baseline, probability distributions due to streamflow and temperature were largely overlapping for trout, but stream temperature became limiting in the end-of-century. For all the species excluding chub, the impacts of streamflow and stream temperature were in opposing directions, suggesting that climate change could lead to both positive and negative changes for riparian species. For sucker, toad, and trout there were positive impacts of flow, but negative impacts of stream

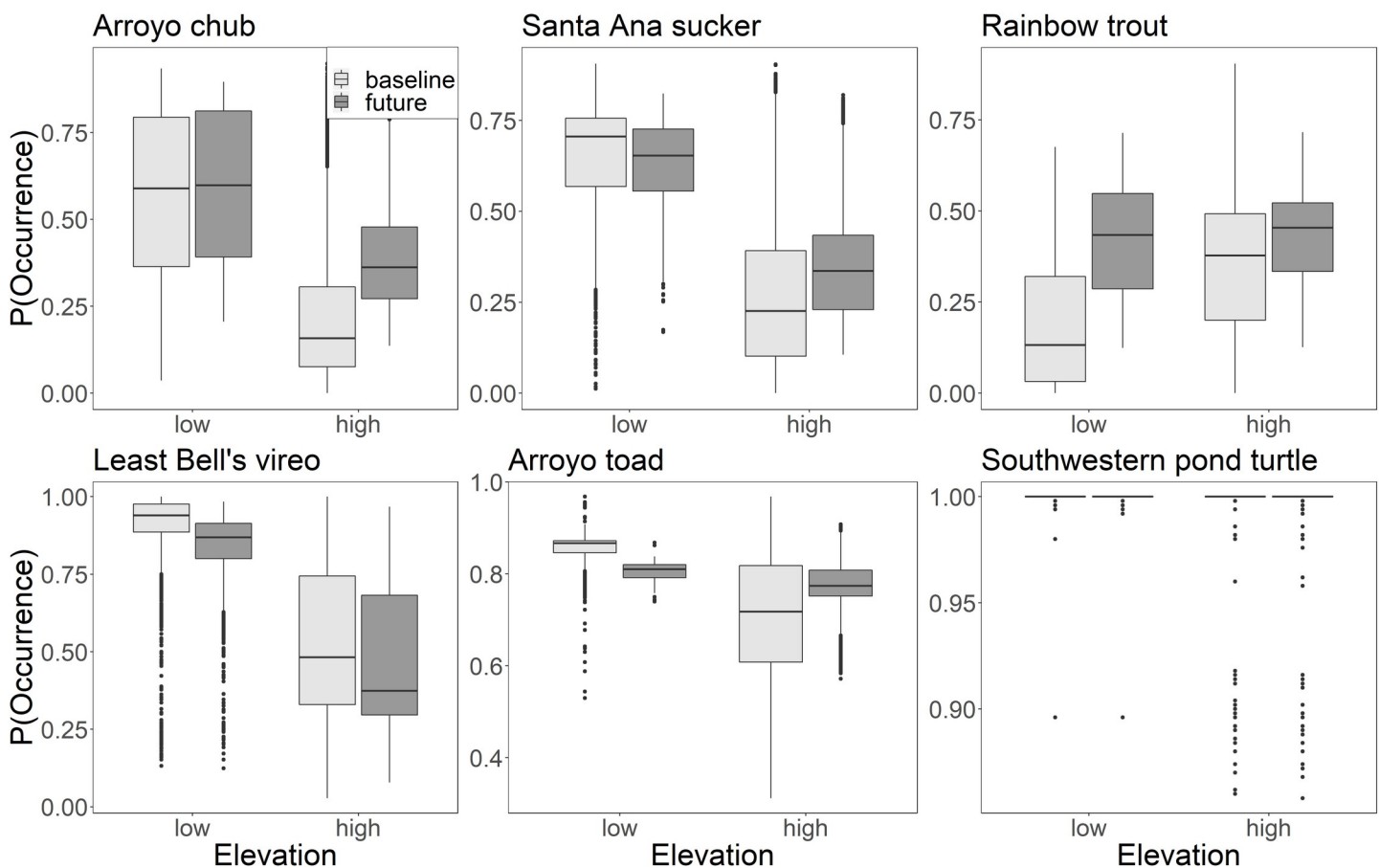

**Fig 4. The distribution of probability of occurrence for the regional stream reaches (including all the stream reaches in the region) due to streamflow in the baseline (light grey) and the end-of-century (dark grey) for low (<375m) and high (>375m) elevation streams.** Table 4 shows the change in the different water year types. Boxplots show the 25th and 75th percentiles, and 1.5*the inter-quartile range at the upper and lower end. Points beyond the 'whiskers' represent outliers.

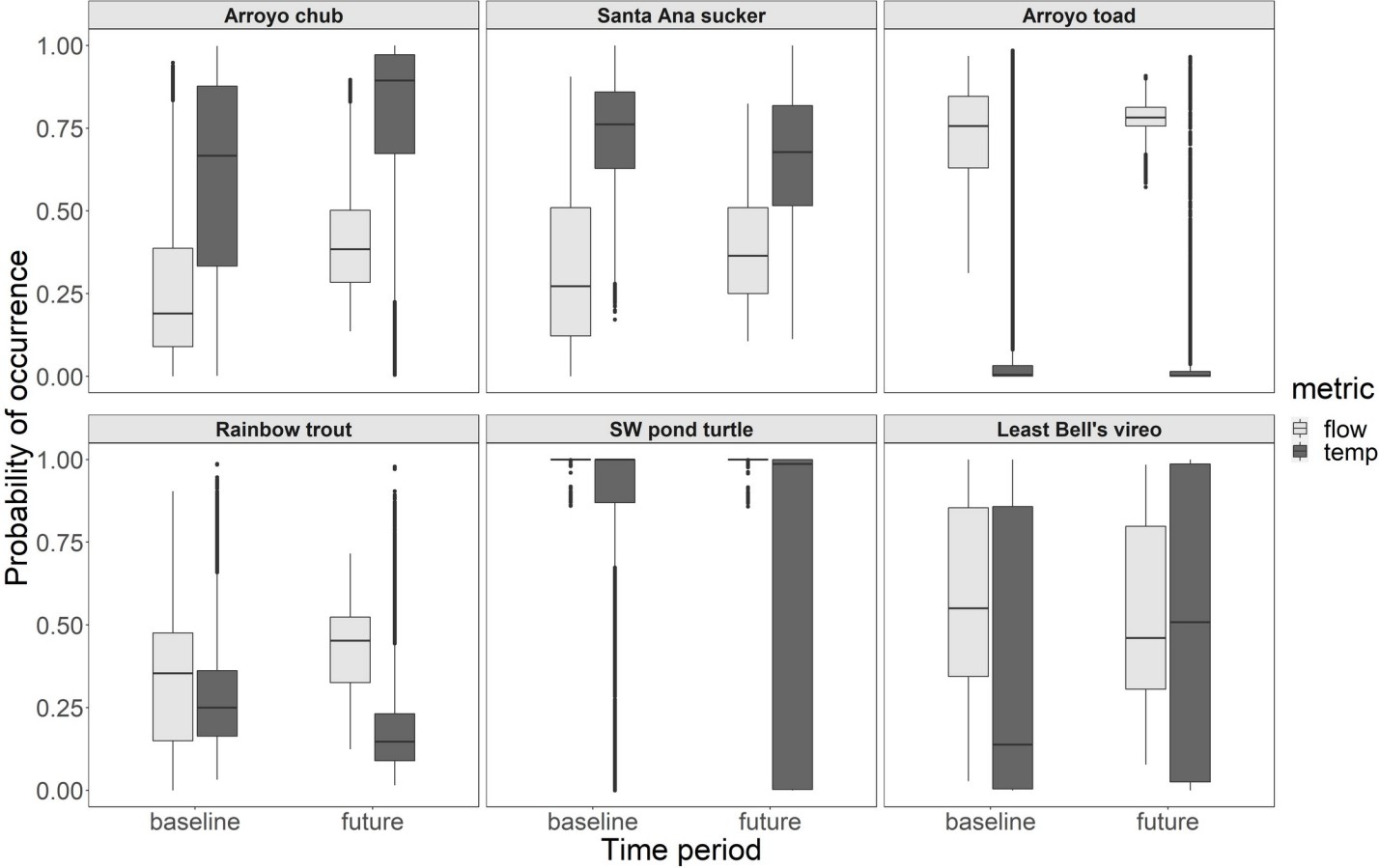

**Fig 5. The distribution of probability of occurrence for the regional stream reaches due to streamflow (light grey) and stream temperature (dark grey) in the baseline and end-of-century time period (future).** In each time period, the lower distribution could be considered the variable that is limiting habitat suitability for each species. Boxplots show the 25th and 75th percentiles, and 1.5*the inter-quartile range at the upper and lower end. Points beyond the 'whiskers' represent outliers.

temperature. For vireo, there were negative impacts of flow, but positive impacts of temperature. Finally, though there was no impact of changing streamflow regime on habitat suitability for the turtle, there were negative impacts of temperature throughout the study region.

## Temporally explicit predictor variables

The 3-year predictor model had an error rate less than or equal to the error rate for the all-year predictor model (Table 5). Validation data accuracy was the same between the two models

**Table 5. Comparison of the two random forest models that used the all-year vs 3-year predictor variables.**

|  | steelhead/rainbow trout |  | Least Bell's vireo |  | Arroyo toad |  | Arroyo chub |  | Santa Ana sucker |  |
| --- | --- | --- | --- | --- | --- | --- | --- | --- | --- | --- |
| **Timeframe** | **All** | **3** | **All** | **3** | **All** | **3** | **All** | **3** | **All** | **3** |
| Validation data accuracy (%) | 0.89 | 0.94 | 1.0 | 1.0 | 0.63 | 0.63 | 0.97 | 0.92 | 0.97 | 0.97 |
| Error rate (%) | 11.06 | 7.53 | 8.16 | 8.16 | 28.0 | 24.0 | 15.17 | 12.92 | 5.91 | 4.30 |

Timeframe refers to the model that was used. "All" means long-term averaged streamflow metrics were used in the model using the entire duration of streamflow timeseries. "Three" means that only short-term streamflow data was used in training the model. For example, only 3 years of streamflow data, that terminated on the date of species observation, were used to calculate the streamflow metrics. The validation data accuracy is the percentage of times the model predicted correctly on the testing dataset. Error rate (out-of-bag estimate of error rate) is the percentage of observations that were misclassified in the training dataset.

except for two species, chub and trout. Probability of occurrence predictions differed between the two models (Fig 6). The temporally explicit model predicted higher probability of occurrence across the study region for chub, sucker, and trout, but lower probability of occurrence for the toad and vireo. In general, the relative predicted species distribution across the region was similar between the models, despite the difference in magnitudes, except for the toad where the temporally explicit model interestingly predicted lower probability of occurrence in the high elevation streams compared to the low elevation streams, whereas the all-year model predicted higher probabilities in the high elevation streams compared to the low elevation streams.

## Discussion

### Species vulnerability patterns

Two characteristics emerged as potential drivers of vulnerability that could be applied regionally to other species in the clusters: Species that occur in high elevation stream habitat and those with a small environmental niche were more vulnerable to climate change, regardless of the geographical extent of their range.

We observed that species in high elevation streams appear to be vulnerable to climate change, while those that occur in lower elevation streams either benefit or are unaffected by predicted conditions. Alternatively, vulnerable and resilient species occurred laterally across the channel in habitats such as the main channel, shallow edge waters, deep pools, and floodplain. This suggests that longitudinal stream position, which correlates with elevation, may be a useful first assessment of vulnerability.

In response to climate change, the two species that generally occur highest in the watersheds (trout and the toad) are projected to lose suitable habitat due to increasing stream temperatures, which is their limiting variable in future years. Conversely, the species that mostly occur in low elevations (chub and vireo) are projected to gain suitable habitat based on increasing probabilities of occurrence due to their limiting variables–streamflow for chub and stream temperature for vireo. An exception is the turtle, which does occur in low elevation streams but is projected to lose suitable habitat due to increasing stream temperatures. Other studies have found greater reduction in range size and risk of extinction for species in high elevation areas [44, 45]. Thuiller [46] found that that species that occur in cool, high elevation areas are potentially more vulnerable to warming, which supports our finding that trout and the toad are projected to be temperature limited. As temperatures rise, the lower elevational limit increases in altitude [47, 48], decreasing habitat at moderate elevations which historically supported the species. Thuiller [46] found that species in very warm regions are less sensitive to warming and predicted that their range expands. Therefore, a replacement of high elevational species by lower elevational species could occur at the transition between habitat types on the mountains, reducing the suitable habitat range of species occupying the highest elevations.

A decrease in habitat suitability at moderate elevations, which leaves just the highest elevations of a species range suitable, may start a negative feedback loop for additional habitat loss. Rivers are connected from headwaters to estuary and the anadromous life form of trout (steelhead) migrate this length. A loss in suitability at moderate elevations could stifle recolonization by adults migrating up the watershed for spawning. Kaylor [49] found that trout populations in the highest headwaters, unlike the other streams in their study, did not rebound after a drought, which they attributed to lack of connectivity at the upstream and downstream end of the reach. In our study, we cannot determine if the decrease in habitat suitability at the downstream end of a suitable reach would stifle trout migration (in which case connectivity would be impacted), because our trout occurrence data combined spawning and migration

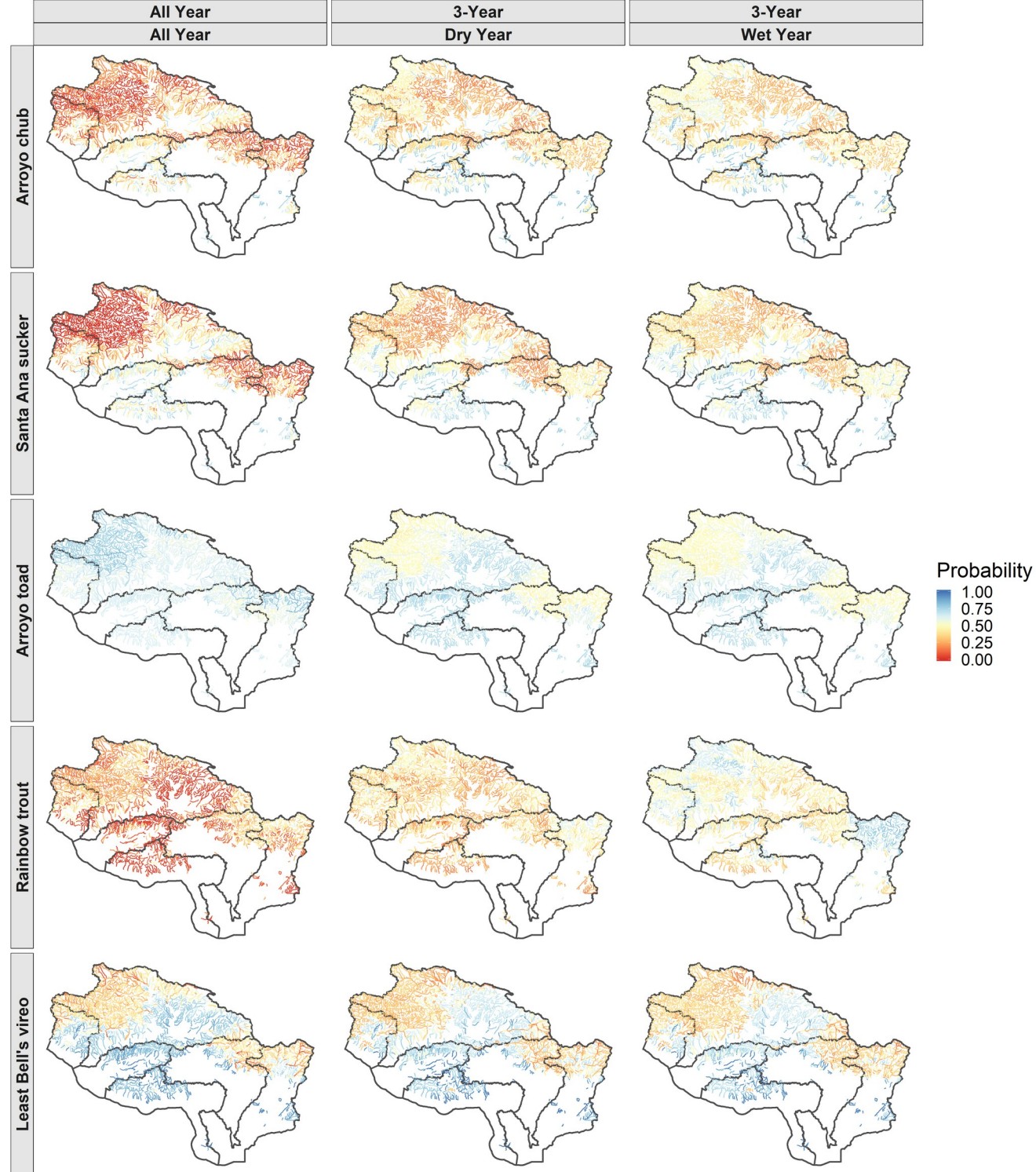

**Fig 6. End-of-century probability of occurrence predictions using the all-year predictors and the 3-year predictors.** The first column shows the predicted end-of-century probabilities using streamflow data from a long-term timeseries including the entire timeseries of forecasted years. The middle column shows predicted end-of-century probability in a three year dry period, and the last column shows predicted end-of-century probability in a three year wet period.

observations. However, it is possible that the highest headwaters become isolated from the lower watersheds, which would stifle recolonization for trout, and potentially the toad depending on their migration ability.

Surprisingly, our results showed that highly endemic species (chub and sucker) were projected to have increased habitat suitability based on their limiting variables, whereas the species with larger ranges throughout western North America (trout and the turtle) were predicted to have decreased habitat suitability. This finding is similar to Thuiller [50] who found endemic plants in Namibia were generally more resilient to climate change than widespread ones. This could suggest that a large geographic range is associated with a negative response to climate change. However, most studies of historic and future climate change have alternatively found that endemic species were vulnerable compared to widespread species (see review: Isaac [51]). Historically, studies of climate warming during the Quaternary found that widespread species were able to inhabit regions of Europe that had previously been too cold but that are currently suitable, whereas range restricted species to this day are limited to the areas that were not frozen for reptiles and amphibians [52] and trees [53]. One explanation for this is that range restricted species are limited by dispersal and cannot move into new territories as they become suitable [54]. This suggests that species with narrow geographic ranges may be more vulnerable than species with wider ranges because the former cannot disperse into new areas and will be sensitive to new climate conditions in the areas they currently inhabit. We did not account for dispersal limitations because our study area is limited to six adjacent watersheds above all major dams, so, in general, dispersal, including the potential for assisted migration, should be feasible. The assumption that any habitat within the region is accessible may be one reason why we did not see greater vulnerability of highly endemic species. Like the historical studies of climate warming, when predicting the relative impact of *future* climate change, Zhang [55] found species with the smallest geographic ranges to be the most vulnerable. Similarly, in a meta-analysis of 131 studies, Urban [1] found that endemic species were more vulnerable to rising temperatures and showed higher extinction risk.

However, perhaps our finding is not contradictory with these other studies of range size and climate change vulnerability. For each focal species, whether the full geographic range is widespread or highly endemic, we project the habitat generalists to respond more favorably than habitat specialists (also observed by Thuiller [46]; Clavel [56]; Poff [5]), which supports a decoupling of geographic range size and environmental niche breadth shown in a study of *Gammarus* spp. [57] and a study of rattlesnakes [58]. While many studies have found a positive association between environmental niche breadth and geographic range [59], in our study, the species with the smallest geographic range often have relatively wide environmental niche breadths. For example, although the toad has a larger geographic range than chub, within its geographic range the toad occupies fewer habitat types. Therefore, despite high endemicity of chub, their projected response to climate change is positive, possibly due to their wide environmental niche breadth within our study region. Overall, the lack of dispersal limitation associated with endemics in this small study area, combined with the wide environmental niche breath, and tolerance of large flow events of the highly endemic species in this study [60, 61], emphasized by the warm stream temperature preferences of chub, could explain why they were less vulnerable to climate change than the more widely dispersed species.

## Species conservation

Future stream temperatures are expected to be warmer throughout the region, and future storm flow magnitudes are projected to be greater and occur more frequently. The lower elevation reaches may retain streamflow for a greater portion of the year, while the high elevation

stream reaches may be drier for a greater portion of the year and experience more rapid recessions. While we modeled the impacts of streamflow and stream temperature separately, the two variables will jointly affect the future habitat suitability for each species.

Projecting the impacts of these changes on the future distributions of riparian species can help managers foresee vulnerabilities for even the less studied species. For example, the regions' two additional native riverine fish, unarmored threespine stickleback (*Gasterosteus aculeatus williamsoni*) and Santa Ana speckled dace (*Rhinichthys osculus* ssp.) occur in the clusters with chub and trout, respectively. Application of our results could suggest that projected streamflow patterns may favor those two species, which occur in low and high elevation stream reaches, respectively. However, increasing stream temperatures likely harm the dace, as extrapolated from the results for trout. Alternatively, rising stream temperatures may support the stickleback, as extrapolated from the results for chub. Vireo represents a cluster of riparian nesting birds, such as yellow warbler (*Setophaga petechia*). Extrapolation of results to these other species would suggest that while projected changes in streamflow will negatively impact their nesting habitat, increases in stream temperature will be supportive. The negative impact of streamflow changes on the vireo cluster may be an increasing problem as the populations of willows that currently provide nesting habitat age. Vireo tend to nest in early successional riparian vegetation that get recruited by the 5- to 10-year storm event [62]. The increased flow magnitudes and frequencies of storm events combined with higher hydroperiods in the low elevation streams where vireo occur could prevent seedlings from recruiting and the result may be an aging population of willows along the streams that currently support nesting. While using the focal species approach is not a complete substitute for species or community level study [63], it can help identify vulnerabilities and inform the direction of future studies.

Habitat suitability modeling that considers the dynamic trends of species distributions improved habitat prioritization for conservation planning [64]. We found that considering temporally specific environmental drivers also produced different results from the model that used long-term averages as predictor variables. For most species the locations of high and low habitat suitability did not differ between the two models despite differences in magnitude, but for the toad the highest probability locations were reversed depending on the model. This suggests that habitat suitability assessment that considers interannual variation can identify conditions that support species that would be missed by using only regional averages, particularly in dynamic habitats like streams in Mediterranean climate regions.

The differences for arroyo toad, compared to the other focal species, could have been magnified because that species in particular limits usage of the stream habitat to years that are suitable. While drought reduces the number of breeding pairs [65], Miller [66] found that alternating drought and storms supported arroyo toad because it can limit breeding to years with suitable conditions while excluding predators that require permanent surface water, such as the American bullfrog (*Lithobates catesbeianus*). However, drought conditions beyond the lifespan of arroyo toad, 6–8 years, impedes breeding opportunities and leads to significant reductions in the population [67]. Temporally specific metrics can reveal how surface water that is present frequently enough will support breeding, but prevent predators.

With these projections, conservation projects can be planned proactively [68]. For example, enhancing stream connectivity between high and low elevation populations will allow the lower population to migrate before conditions become unsuitable. Additionally, the spatial prioritization of location and types of land parcels for protection [69], decisions regarding a species vulnerability and sensitivity listing, and water management decisions can be informed by predicting spatial habitat suitability. Projections of where suitable habitat conditions may persist can help managers target conservation and restoration projects in watersheds that maintain those favorable conditions [70]. For example, managers can begin monitoring in locations projected to support species

to assess for the presence of other stressors and opportunities to enhance the habitat such as riparian tree planting programs to buffer against rising air temperatures.

## Consideration for futures studies

There are other drivers of species distribution besides streamflow and stream temperature that may change with climate change, such as biotic competition [71], barriers to dispersal [71, 72], and land use [39]. We did not include the entire range of some of the species which underestimates the range of environmental tolerances [73]. This may underestimate impacts of climate change for species toward the southern end of their range (trout and the turtle) and potentially overestimate impacts of climate change for species at the mid- to northern end of their range (arroyo toad). Similarly, climate change impacts could affect vireo in other parts of their range or during migration that we could not consider [74].

Another consideration would be to model different life phases of the focal species separately. We combined the life history forms of trout, and all aquatic life history stages of each focal species. We did this so we could include survey data when life history stage was not reported. We therefore lumped all reports of either life history form of trout and all life history stages of each focal species into the presence and absence data set, even though life history stages are associated with different environmental conditions [49, 75, 76] because we judged data representing as much of the study region as possible to be more important. The magnitude of the projected impact on trout was likely diminished because we included migration observations in the species occurrence data set from low elevations, as opposed to only including observations from their spawning grounds in high elevation tributaries.

We assumed full migration capacity of all species in the future scenarios, meaning any new suitable habitat is considered regardless of proximity to the current range. Other studies have used two extreme dispersal scenarios, full or none, to determine future suitable habitat [44, 77, 78], however, we considered stream reaches that would be suitable for species occurrence to give wildlife managers flexibility for conservation planning. Forecasts of suitable stream reaches that are separated from the current range by natural or anthropogenic barriers can help direct assisted migration programs or connectivity design [79] and in some cases, a separate watershed may provide better refuge due to practical reasons such as less recreational use.

Finally, analyses of this type are heavily dependent on the air temperature and precipitation predictions that drive the stream habitat modeling and ultimately the habitat suitability toward particular organisms. While predictions from global climate models agree that air temperature will increase [80], there is far more uncertainty in the future precipitation predictions [16, 36, 81]- some models predict an increase in total annual precipitation while others predict a decrease. However, regardless of the direction of precipitation change, some streamflow predictions, like an increase in large flood events, are consistent across models [16, 82, 83]. For example, of the three GCMs used in this analysis, MIROC5 predicts an overall decrease in precipitation, CCSM4 predicts close to no change in precipitation, and CanESM2 predicts an increase in total precipitation, yet all three were used in rainfall-runoff models that ultimately predicted increases in stormflow magnitude and frequency yet a decrease in hydroperiod [34]. Therefore, we do not expect the high uncertainty of future precipitation to impact our results greatly, but future studies should continue to use the newest models that capture the range of uncertainty in case discrepancies in streamflow prediction between models become apparent.

## Conclusion

Here we have demonstrated an approach for projecting the impacts of climate change on riparian species due to changes in streamflow and stream temperature. By using temporally

and spatially referenced streamflow and stream temperature metrics that align with species occurrence, models more accurately describe the conditions associated with species occurrence, which can help managers. We recommend investigating the importance of short-term stream conditions for species that use the stream selectively to better model habitat suitability. By using species that represented different portions of the watershed, we can estimate impacts on other species with similar habitat preferences. We find that while projections differed between species, there were some patterns that could be applied elsewhere to support an initial estimate at species vulnerabilities. Species in the high elevation streams are likely more vulnerable to climate change. Finally, we recommend this method to provide a preliminary estimation of the biological impacts of climate change that can be broadly applied to other species in the region.

## Supporting information

**S1 File.**
(DOCX)

## Acknowledgments

We thank Alex Hall, Jerry H-Y. Huang, and Neil Berg, at the University of California, Los Angeles, Department of Atmospheric and Oceanic Sciences, for providing their downscaled air temperature and precipitation model output for baseline conditions and future projections. We thank Rosi Dagit from the Resource Conservation District of Santa Monica Mountains, Mary Larson and Jeff Weaver from California Department of Fish and Wildlife, Chris Medak and Chris Dellith from the United States Fish and Wildlife Service, and Johnathan Baskin from California State Polytechnic University Pomona for providing species occurrence data. We thank Kelly Flint and Alicia Kinoshita from the Department of Civil Engineering, San Diego State University for modeling streamflow. Finally, we thank members of our technical advisory committee for their thoughtful comments throughout the study.

## Author Contributions

**Conceptualization:** Eric D. Stein.

**Data curation:** Jennifer B. Rogers, Marcus W. Beck.

**Formal analysis:** Jennifer B. Rogers, Marcus W. Beck.

**Funding acquisition:** Eric D. Stein.

**Investigation:** Jennifer B. Rogers.

**Methodology:** Jennifer B. Rogers, Eric D. Stein, Marcus W. Beck.

**Project administration:** Eric D. Stein.

**Supervision:** Richard F. Ambrose.

**Validation:** Jennifer B. Rogers, Richard F. Ambrose.

**Visualization:** Jennifer B. Rogers, Marcus W. Beck.

**Writing – original draft:** Jennifer B. Rogers.

**Writing – review & editing:** Jennifer B. Rogers, Eric D. Stein, Marcus W. Beck, Richard F. Ambrose.

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
