## [Decision Letter · Decision Letter 0]

27 Aug 2020

PONE-D-20-20248

The impact of climate change induced alterations of streamflow and stream temperature on the distribution of riparian species

PLOS ONE

Dear Dr. Rogers,

Thank you for submitting your manuscript to PLOS ONE. After careful consideration, we feel that it has merit but does not fully meet PLOS ONE’s publication criteria as it currently stands. Therefore, we invite you to submit a revised version of the manuscript that addresses the points raised during the review process.

We look forward to receiving your revised manuscript.

Kind regards,

Julia A. Jones

Academic Editor

PLOS ONE

Journal Requirements:

2. We noted in your submission details that a portion of your manuscript may have been presented or published elsewhere. [An earlier version of this analysis has been published in the UCLA doctoral dissertation of the first author.] Please clarify whether this [conference proceeding or publication] was peer-reviewed and formally published. If this work was previously peer-reviewed and published, in the cover letter please provide the reason that this work does not constitute dual publication and should be included in the current manuscript.

3.We note that you have indicated that data from this study are available upon request. PLOS only allows data to be available upon request if there are legal or ethical restrictions on sharing data publicly. For more information on unacceptable data access restrictions, please see http://journals.plos.org/plosone/s/data-availability#loc-unacceptable-data-access-restrictions.

4.We note that [Figure(s) 1, 3 and 6] in your submission contain [map/satellite] images which may be copyrighted. All PLOS content is published under the Creative Commons Attribution License (CC BY 4.0), which means that the manuscript, images, and Supporting Information files will be freely available online, and any third party is permitted to access, download, copy, distribute, and use these materials in any way, even commercially, with proper attribution. For these reasons, we cannot publish previously copyrighted maps or satellite images created using proprietary data, such as Google software (Google Maps, Street View, and Earth). For more information, see our copyright guidelines: http://journals.plos.org/plosone/s/licenses-and-copyright.

1.    You may seek permission from the original copyright holder of Figure(s) [1, 3 and 6] to publish the content specifically under the CC BY 4.0 license. 

Reviewers' comments:

Reviewer's Responses to Questions

**Comments to the Author**

1. Is the manuscript technically sound, and do the data support the conclusions?

Reviewer #1: Yes

2. Has the statistical analysis been performed appropriately and rigorously? 

Reviewer #1: Yes

3. Have the authors made all data underlying the findings in their manuscript fully available?

Reviewer #1: Yes

4. Is the manuscript presented in an intelligible fashion and written in standard English?

Reviewer #1: Yes

5. Review Comments to the Author

Reviewer #1: This article presents an analysis of the impact of climate change on aquatic species in stream ecosystems, with a focus on the impacts of flow regime and water temperature. The authors model the occurrence of 6 representative or "focal" species as a function of hydrologic and thermal variables, and forecast species ranges under different future climate scenarios. They conclude that high-altitude species will experience range contraction, while low-elevation species will experience range expansion. Overall, the paper is well-written, the analysis comprehensive and well executed, and the results of broad potential interest. The compilation of species distributions is particularly impressive. The work could be published with minor revisions, pending clarification of some methods and a more detailed interpretation of the relationship between flow metrics, how they change under future climate, and their role in species distribution.

Please further clarify the methods used for the reach-scale predictions of hydrologic metric...you have continuous daily flow observations and/or HEC-HMS model for all reaches. Where you have observations of species, how do you estimate hydrologic metrics for 2090s? Where you have the HEC model only, do you use HEC results to "downscale" from a lumped watershed scale, or does HEC produce continuous flow predictions for all reaches in a given watershed?

I would have liked some discussion of how the flow metrics relate to species presence/absence, how flow changes between the basline and future scenarios, and how those two helps interpret the results from the species distribution model. In general, do the predictive flow metrics and their sign in the model (as presented in Table S-4) make biological sense (e.g. fish presence increases with more flow, etc)? Which key flow metrics change, and what is the direction of change? I think this might help to come up with a more well-rounded story, e.g. “Future climates may have warmer water but also flashier (?) hydrographs and lower baseflow (?) which favor/disfavor xx species / functional types.”

The reliability of the flow projections and impacts on species depend on the accuracy of the future precipitation, which is notoriously difficult to predict. Are your results likely to be sensitive to changes in simulated precipitation—maybe not if temperature was the most important factor for future species occurrence. There is no need to quantify the impact of precipitation uncertainty, just a few sentences acknowledging it and suggesting how different types of error (over, under-estimation) may impact or not impact your results.

Minor comments:

The main text and tables use the Latin names for the study species, while the Figures use the common names. Personally I would prefer that the common names be used throughout (as it makes it easier for the non-biologist to remember and associate results with the species), with the Latin names included in one location for establishing the correspondence between common and Latin names.

The color scheme in Fig 3 is different for each species. This is ok, but it complicates the visual comparison of all species together. Did you explore using a common scale for all, where e.g. red means increase and blue means decrease?

In Fig 3, would it help to put all descreasing species on the one side (e.g. left) and increasing spp on the other side? That would facilitate comparison of the boxplots and highlight similarities of increasing/decreasing species.

In Fig 3, Rainbow Trout shows negative delta P (lower P occurrence in future given changes in stream temperature), but in Figure 4, it P(occurrence) seems to increase in future climates due to streamflow changes. I think you could highlight this more in the results: The effects of streamflow changes counteracted the effects of temperature changes for some (all?) species.

L379-389: Fig 4 is confusing to me, especially interpretation of the baseline simulations…you run P(occurrence) models for baseline scenario with either flow or temperature as the predictor. But for the basline you have the actual P(occurrence) from the data, so is it possible to put that on the graph as a dotted horizontal line? Or maybe this is for a wider geographic area than the training and validation data?

L391-401. Fig 6 shows very different P(occurrence) for several species between the All and either Dry or Wet year simulations, yet the error is the same. How is that possible? Which one is best for modeling future scenarios? It’s possible you say this elsewhere, but it would be worth stating here for clarity.

Table S-4 caption says that the variables used in the analysis in bold, but no rows in Table S-4 are in bold.

Table S-4 caption “, from the year of analysis”. Year of analysis could be the year that the observation of spp presence or absence, yes? If so, please add to the caption.

Please also see the annotated PDF, which has comments and grammatical suggestions.

6. PLOS authors have the option to publish the peer review history of their article (what does this mean?). If published, this will include your full peer review and any attached files.

Reviewer #1: **Yes: **Trent W. Biggs

---

## [Author Response · Author response to Decision Letter 0]

27 Oct 2020

Academic Editor Comments

We corrected the heading levels: Level 1 for major sections and level 2 for sub-sections

We corrected the format of the in-text citations for supporting material

We corrected the format of the author affiliations, removed the running header, and centered the article title.

2. We noted in your submission details that a portion of your manuscript may have been presented or published elsewhere. [An earlier version of this analysis has been published in the UCLA doctoral dissertation of the first author.] Please clarify whether this [conference proceeding or publication] was peer-reviewed and formally published. If this work was previously peer-reviewed and published, in the cover letter please provide the reason that this work does not constitute dual publication and should be included in the current manuscript.

The dissertation was not peer reviewed and was not published as a conference proceeding or publication. It is available for download (https://escholarship.org/uc/item/1d13q6zk), however, we would like to clarify that it has not been formally published elsewhere. The submitted version to PLOS ONE is similar to chapter 3, but with substantial revisions as needed for the primarily literature and following reviewer comments. 

3.We note that you have indicated that data from this study are available upon request. PLOS only allows data to be available upon request if there are legal or ethical restrictions on sharing data publicly. For more information on unacceptable data access restrictions, please see http://journals.plos.org/plosone/s/data-availability#loc-unacceptable-data-access-restrictions.

 The data that we originally excluded from the Zenodo repository was the species occurrence data, because much of that data was provided to us by third party. However, the individuals who did provide data to us have been contacted and have each responded that it is okay to include the data on the website at the spatial scale used in the analysis, which is the NHD reach. The species occurrence data is now in the GitHub repository and the Zenodo repository listed in the data availability statement.

4.We note that [Figure(s) 1, 3 and 6] in your submission contain [map/satellite] images which may be copyrighted. All PLOS content is published under the Creative Commons Attribution License (CC BY 4.0), which means that the manuscript, images, and Supporting Information files will be freely available online, and any third party is permitted to access, download, copy, distribute, and use these materials in any way, even commercially, with proper attribution. For these reasons, we cannot publish previously copyrighted maps or satellite images created using proprietary data, such as Google software (Google Maps, Street View, and Earth). For more information, see our copyright guidelines: http://journals.plos.org/plosone/s/licenses-and-copyright.

We believe that shapefiles that you are referring to include the USGS National Hydrography Dataset flow lines, the watershed boundaries, and the outline of California. These three files are freely available for public use and are not copyrighted. We did add an actual citation for the NHD data (before we simply put the website in parentheses). 

  

Reviewer Comments

5. Review Comments to the Author

Reviewer #1: This article presents an analysis of the impact of climate change on aquatic species in stream ecosystems, with a focus on the impacts of flow regime and water temperature. The authors model the occurrence of 6 representative or "focal" species as a function of hydrologic and thermal variables, and forecast species ranges under different future climate scenarios. They conclude that high-altitude species will experience range contraction, while low-elevation species will experience range expansion. Overall, the paper is well-written, the analysis comprehensive and well executed, and the results of broad potential interest. The compilation of species distributions is particularly impressive. The work could be published with minor revisions, pending clarification of some methods and a more detailed interpretation of the relationship between flow metrics, how they change under future climate, and their role in species distribution.

Please further clarify the methods used for the reach-scale predictions of hydrologic metric...you have continuous daily flow observations and/or HEC-HMS model for all reaches. Where you have observations of species, how do you estimate hydrologic metrics for 2090s? Where you have the HEC model only, do you use HEC results to "downscale" from a lumped watershed scale, or does HEC produce continuous flow predictions for all reaches in a given watershed?

This is a great suggestion. The reason for our brevity is because we have a separate manuscript where we describe the hydrologic and stream temperature modeling approach, which is currently in review at Ecohydrology. However, we agree that we were too brief in our description in this manuscript, so we added in description of our method, without completely rewriting information from the other manuscript. Here is the section with the additional explanation to describe the reach scale predictions of the hydrologic metrics, which include the HEC-HMS modeling and the random forest estimation to the remaining stream reaches in the baseline and end-of-century:

“Daily flow time series were compiled from a combination of flow gages and the U.S. Army Corps of Engineers Hydrologic Engineering Center Hydrologic Modeling System (HEC-HMS) rainfall-runoff model of a subset of watersheds using modeled precipitation data (33). In total, 68 watersheds were selected to represent our study region and modeled using HEC-HMS with watershed characteristics designated using a method developed by Sengupta et al. (2018)(34). The flow gauges and the downstream terminus of the 68 watersheds were in locations where biological surveys have occurred for the six focal species so that the monitored or modeled flow could be associated with biological condition.

Precipitation for the HEC-HMS models were from a 90-meter, gridded precipitation data set, which consisted of a continuous time series spanning water years 1982-2014 at a 3-hourly time step across the study region (Berg et al., 2015; downscaled by Huang & Hall, 2018). Each HEC-HMS model was run at a 3-hourly time-step and then averaged into daily average flow values, resulting in a daily streamflow time series spanning water years 1982-2014 for each watershed.

To build habitat suitability models (see section below on Biological Modeling) hydrologic metrics (S-4) were calculated from the flow time series (from either the gauge or the model) for each species’ presence or absence record at the time and location of the biological survey. Each metric was calculated for short periods from when a species occurrence was observed (3-, 5-, and 10-years prior to the observation) and for the entire duration of the timeseries. To predict habitat suitability throughout the watershed, the hydrologic metrics were estimated regionally (i.e. for every stream reach) using physical basin characteristics and precipitation (33). Briefly, with the hydrologic metrics from the 68 watersheds and gauged reaches, we used random forest in R (37) to predict streamflow metrics for all NHD reaches in the study for the baseline wet, dry and moderate periods. We used watershed characteristics from the EPA StreamCat database (18), such as elevation and area, and precipitation metrics, such as ‘number of storm events’, which were derived from the baseline precipitation data. 

To predict the flow metrics for the three end-of-century time periods, the random forest model was applied with the same watershed characteristic predictors, but with the end-of-century precipitation data for the wet, dry, and moderate periods. Some watershed characteristics like land cover change can impact stream habitat (38); however, because our region includes only the unaltered watersheds predominantly in mountainous areas, many of which are protected, we do not anticipate substantial urbanization in future years. All processing for the streamflow metrics was completed in R (39). Ultimately, we modeled metrics for each NHD stream reach for the baseline and end-of-century years. End-of-century streamflow metrics were averaged across the three GCMs.” 

I would have liked some discussion of how the flow metrics relate to species presence/absence, how flow changes between the basline and future scenarios, and how those two helps interpret the results from the species distribution model. In general, do the predictive flow metrics and their sign in the model (as presented in Table S-4) make biological sense (e.g. fish presence increases with more flow, etc)? Which key flow metrics change, and what is the direction of change? I think this might help to come up with a more well-rounded story, e.g. “Future climates may have warmer water but also flashier (?) hydrographs and lower baseflow (?) which favor/disfavor xx species / functional types.”

We added additional discussion to the Species Conservation section of the discussion. This is a really important comment, but of course difficult because it involves making assumptions about which variable, flow or temperature, might be more or less important. 

We added text to give a synopsis to how flow and temperature change between baseline and future years and this serves as a concrete backdrop for the next paragraph when we discuss how flow changes might be expected to impact the trout and chub clusters:

“Future stream habitats are expected to be warmer throughout the entire region, and future storm flow magnitudes are projected to be greater and occur more frequently. The lower elevation reaches may retain streamflow for a greater portion of the year, while the high elevation stream reaches may be drier for a greater portion of the year and experience more rapid recessions. While we assessed the impacts of streamflow and stream temperature separately, the two variables will jointly affect the future habitat suitability for each species.”

We also added text to discuss how we hypothesize the flow changes may impact vireo:

“The negative impact of streamflow changes on the vireo cluster may be an increasing problem as the populations of willows that currently provide nesting habitat age. Vireo tend to nest in early successional riparian vegetation that get recruited by the 5- to 10-year storm event (59). The increased flow magnitudes and frequencies of storm events combined with higher hydroperiods in the low elevation streams where vireo occur could prevent seedlings from recruiting and the result may be an aging population of willows along the streams that currently support nesting.“

The reliability of the flow projections and impacts on species depend on the accuracy of the future precipitation, which is notoriously difficult to predict. Are your results likely to be sensitive to changes in simulated precipitation—maybe not if temperature was the most important factor for future species occurrence. There is no need to quantify the impact of precipitation uncertainty, just a few sentences acknowledging it and suggesting how different types of error (over, under-estimation) may impact or not impact your results.

Thank you for this recommendation. We fully appreciate that the results of this research are rooted in future climate predictions that are uncertain. We added in a paragraph to the Consideration for Future Studies section of the discussion to acknowledge the uncertainty and give our suggestion of its importance to this analysis. 

“Finally, analyses of this type are heavily dependent on the air temperature and precipitation predictions that drive the stream habitat modeling and ultimately the habitat suitability toward particular organisms. While predictions from global climate models agree that air temperature will increase (77), there is far more uncertainty in the future precipitation predictions (16,35,78)- some models predict an increase in total annual precipitation while others predict a decrease. However, regardless of the direction of precipitation change, some streamflow predictions, like an increase in large flood events, are consistent across models (16,79,80). For example, of the three GCMs used in this analysis, MIROC5 predicts an overall decrease in precipitation, CCSM4 predicts close to no change in precipitation, and CanESM2 predicts an increase in total precipitation, yet all three were used in rainfall-runoff models that ultimately predicted increases in stormflow magnitude and frequency yet a decrease in hydroperiod (33). Therefore, we do not expect the high uncertainty of future precipitation to impact our results greatly, but future studies should continue to use the newest models that capture the range of uncertainty in case discrepancies in streamflow prediction between models become apparent.”

Minor comments:

The main text and tables use the Latin names for the study species, while the Figures use the common names. Personally I would prefer that the common names be used throughout (as it makes it easier for the non-biologist to remember and associate results with the species), with the Latin names included in one location for establishing the correspondence between common and Latin names.

We changed all the names in the main text to an abbreviated common name (for example, trout, instead of southern California steelhead/rainbow trout. In tables and figures we use the full common name as space allow. We agree that it makes it easier for those not familiar with these particular species. The first mention of the species name still has the full common name along with the full scientific name, and we added in the abbreviation used in the remainder of the document.

The color scheme in Fig 3 is different for each species. This is ok, but it complicates the visual comparison of all species together. Did you explore using a common scale for all, where e.g. red means increase and blue means decrease?

Thank you for this feedback. We would have liked to have the range be the same for each species (-1 to 1) but that made it hard to see the contrast throughout the study region where the actual range of change values was much less than the total range of the color scale. We decided to make the mid-point of each panel 0 so that, as you suggested, red represents a decrease and blue represents an increase. Though the range of values differs between panels, now the colors are the same, which would make it easier for someone to scan the figure quickly and see general increasing or decreasing trends.

In Fig 3, would it help to put all descreasing species on the one side (e.g. left) and increasing spp on the other side? That would facilitate comparison of the boxplots and highlight similarities of increasing/decreasing species.

Good suggestion. With the new temperature model, only two species were decreasing, so we put them in the top row for easy comparison.

In Fig 3, Rainbow Trout shows negative delta P (lower P occurrence in future given changes in stream temperature), but in Figure 4, it P(occurrence) seems to increase in future climates due to streamflow changes. I think you could highlight this more in the results: The effects of streamflow changes counteracted the effects of temperature changes for some (all?) species.

We added a short paragraph to the Impacts of stream temperature versus streamflow section of the results. 

“For all the species excluding chub, the impacts of streamflow and stream temperature were in opposing directions, suggesting that climate change could lead to both positive and negative changes for riparian species. For sucker, toad, and trout there were positive impacts of flow, but negative impacts of stream temperature. For vireo, there were negative impacts of flow, but positive impacts of temperature. Finally, though there was no impact of changing streamflow regime on habitat suitability for the turtle, there were negative impacts of temperature throughout the study region.”

L379-389: Fig 4 is confusing to me, especially interpretation of the baseline simulations…you run P(occurrence) models for baseline scenario with either flow or temperature as the predictor. But for the basline you have the actual P(occurrence) from the data, so is it possible to put that on the graph as a dotted horizontal line? Or maybe this is for a wider geographic area than the training and validation data?

Figure 4 is just showing the probability of occurrence that results from using streamflow as the predictor variables (Fig 3 shows something similar for stream temperature). The data for each box and whisker plot is the stream reaches in either low or high elevations. The baseline shows the predicted probabilities for all of the stream reaches in the region. Even though it is baseline, it is still predicted because we do not know presence and absence for every stream reach. Instead, we just know presence or absence in a few stream reaches (where surveys have occurred in the past few decades), which is what we used to build the habitat suitability model. Then, with the streamflow metrics in the remainder of the watershed, we can predict probability of occurrence. There is no actual or observed P(occurrence) in the baseline that we could represent with a single dotted line, because the actual observations are simply ‘presence’ or ‘absence’ at a particular reach. Your last sentence is correct, this is a wider geographic area than the training/validation data (which were limited to the historical observations), it includes every stream reach in the study region – we added this into the figure caption to clarify.

L391-401. Fig 6 shows very different P(occurrence) for several species between the All and either Dry or Wet year simulations, yet the error is the same. How is that possible? Which one is best for modeling future scenarios? It’s possible you say this elsewhere, but it would be worth stating here for clarity.

The values given in Table 5 report the model performance on the training and validation data, whereas the values in Figure 6 show the end of century predictions. The models do predict quite different futures. However, the random forest process makes 500 predictions of “present” or “absent”. The probability is the percentage of “present” out of that 500. The accuracy and error rate instead represent the entire 500 trees and if more than 50% were “present”, in which case it gets a “present” and this gets compared to the actual observation to determine accuracy. Therefore, the probability can vary greatly while the accuracy stays the same. Therefore, even though the predictions from the two models look different, they are not so different that they would actually have had different (or substantially different) model performance metrics. 

Table S-4 caption says that the variables used in the analysis in bold, but no rows in Table S-4 are in bold.

This was a typo – we removed that sentence from the caption.

Table S-4 caption “, from the year of analysis”. Year of analysis could be the year that the observation of spp 

presence or absence, yes? If so, please add to the caption.

We added this in to clarify “(either the year the species was observed or the wet/dry/moderate year of interest)”

Please also see the annotated PDF, which has comments and grammatical suggestions.

Thank you for the revisions on grammar and sentence structure. Changes were made throughout the document.

---

## [Editor Report · Decision Letter 1]

9 Nov 2020

The impact of climate change induced alterations of streamflow and stream temperature on the distribution of riparian species

PONE-D-20-20248R1

Dear Dr. Rogers,

We’re pleased to inform you that your manuscript has been judged scientifically suitable for publication and will be formally accepted for publication once it meets all outstanding technical requirements.

Kind regards,

Julia A. Jones

Academic Editor

PLOS ONE
---

## [Editor Report · Acceptance letter]

13 Nov 2020

PONE-D-20-20248R1 

The impact of climate change induced alterations of streamflow and stream temperature on the distribution of riparian species 

Dear Dr. Rogers:

I'm pleased to inform you that your manuscript has been deemed suitable for publication in PLOS ONE. Congratulations! Your manuscript is now with our production department. 

Kind regards, 

on behalf of

Dr. Julia A. Jones 

Academic Editor

PLOS ONE